# Testing approaches to sharing trial results with participants: The Show RESPECT cluster randomised, factorial, mixed methods trial

**Annabelle South**[1]*, **Nalinie Joharatnam-Hogan**[1], **Cara Purvis**[1], **Elizabeth C. James**[1], **Carlos Diaz-Montana**[1], **William J. Cragg**[2], **Conor Tweed**[1], **Archie Macnair**[1], **Matthew R. Sydes**[1], **Claire Snowdon**[3], **Katie Gillies**[4], **Talia Isaacs**[5], **Barbara E. Bierer**[6], **Andrew J. Copas**[1]

1 MRC CTU at UCL, Institute of Clinical Trials and Methodology, UCL, London, United Kingdom, 2 Clinical Trials Research Unit, Leeds Institute of Clinical Trials Research, University of Leeds, Leeds, United Kingdom, 3 London School of Hygiene and Tropical Medicine, London, United Kingdom, 4 Health Services Research Unit, University of Aberdeen, Aberdeen, United Kingdom, 5 UCL Institute of Education, University College London, London, United Kingdom, 6 Harvard Medical School and Brigham and Women's Hospital, Boston, Massachusetts, United States of America

* a.south@ucl.ac.uk

**Data Availability Statement:** The protocol is available on the MRC CTU website https://www.

## Abstract

### Background

Sharing trial results with participants is an ethical imperative but often does not happen. We tested an Enhanced Webpage versus a Basic Webpage, Mailed Printed Summary versus no Mailed Printed Summary, and Email List Invitation versus no Email List Invitation to see which approach resulted in the highest patient satisfaction with how the results were communicated.

### Methods and findings

We carried out a cluster randomised, 2 by 2 by 2 factorial, nonblinded study within a trial, with semistructured qualitative interviews with some patients (ISRCTN96189403). Each cluster was a UK hospital participating in the ICON8 ovarian cancer trial. Interventions were shared with 384 ICON8 participants who were alive and considered well enough to be contacted, at 43 hospitals. Hospitals were allocated to share results with participants through one of the 8 intervention combinations based on random permutation within blocks of 8, stratified by number of participants. All interventions contained a written plain English summary of the results. The Enhanced Webpage also contained a short video. Both the Enhanced Webpage and Email contained links to further information and support. The Mailed Printed Summary was opt-out.

Follow-up questionnaires were sent 1 month after patients had been offered the interventions. Patients' reported satisfaction was measured using a 5-point scale, analysed by ordinal logistic regression estimating main effects for all 3 interventions, with random effects for site, restricted to those who reported receiving the results and assuming no interaction. Data collection took place in 2018 to 2019.

ctu.mrc.ac.uk/media/1980/show-respect_protocol_v30_20aug2018_clean.pdf. The individual participant data, qualitative and quantitative, that underlie the results reported in this Article, after de-identification, will be available following the MRCCTU's standard moderated access approach (details of which are available https://www.ctu.mrc.ac.uk/our-research/other-research-policy/data-sharing/). Some of the data used in the analyses presented in this paper were obtained from the underpinning ICON8 trial, so permission to share these specific data items (age, ICON8 arm, ICON8 participant number) would also be required from the ICON8 Trial Steering Committee. Applicants will need to state the aims of any analyses and provide a methodologically sound proposal. Applications should be directed to mrcctu.datareleaserequest@ucl.ac.uk. Data requestors will need to sign a data access agreement at an institutional level.

**Funding:** The Show RESPECT study was funded by the Medical Research Council through core grants to MRS at the MRC CTU at UCL for Trial Conduct Methodology (MC_UU12023/24 and MC_UU_00004/08) https://mrc.ukri.org/. The funder had no role in the study design, the collection, analysis and interpretation of data, the writing of the report and the decision to submit the article for publication. All authors had full access to the study data, including statistical reports and tables, and can take responsibility for the integrity of the data and the accuracy of the data analysis.

**Competing interests:** I have read the journal's policy and the authors of this manuscript have the following competing interests: AS, ECJ, CP, CT, CS, AM, AJC, NJH, CDM, KG, TI, BEB and WJC have nothing to declare. MRS reports grants and non-financial support from Astellas, grants from Clovis, grants and non-financial support from Janssen, grants and non-financial support from Novartis, grants and non-financial support from Pfizer, grants and non-financial support from Sanofi, personal fees from Lilly Oncology, personal fees from Janssen, outside the submitted work.

**Abbreviations:** ICC, Intracluster Correlation Coefficient; ISCM, Information-seeking and Communication Model; mITT, modified intention to treat; OR, odds ratio; PPI, patient and public involvement; Show RESPECT, Show Results to Participants Engaged in Clinical Trials.

Questionnaires were sent to 275/384 randomly selected participants and returned by 180: 90/142 allocated Basic Webpage, 90/133 Enhanced Webpage; 91/141 no Mailed Printed Summary, 89/134 Mailed Printed Summary; 82/129 no Email List Invitation, 98/146 Email List Invitation. Only 3 patients opted out of receiving the Mailed Printed Summary; no patients signed up to the email list. Patients' satisfaction was greater at sites allocated the Mailed Printed Summary, where 65/81 (80%) were quite or very satisfied compared to sites with no Mailed Printed Summary 39/64 (61%), ordinal odds ratio (OR) = 3.15 (1.66 to 5.98, $p$ < 0.001). We found no effect on patient satisfaction from the Enhanced Webpage, OR = 1.47 (0.78 to 2.76, $p$ = 0.235) or Email List Invitation, OR = 1.38 (0.72 to 2.63, $p$ = 0.327). Interviewees described the results as interesting, important, and disappointing (the ICON8 trial found no benefit). Finding out the results made some feel their trial participation had been more worthwhile. Regardless of allocated group, patients who received results generally reported that the information was easy to understand and find, were glad and did not regret finding out the results. The main limitation of our study is the 65% response rate.

## Conclusions

Nearly all respondents wanted to know the results and were glad to receive them. Adding an opt-out Mailed Printed Summary alongside a webpage yielded the highest reported satisfaction. This study provides evidence on how to share results with other similar trial populations. Further research is needed to look at different results scenarios and patient populations.

## Trial registration

ISRCTN: ISRCTN96189403.

---

## Author summary

### Why was this study done?

- Previous research has shown that most people who take part in clinical trials want to be told the results of those trials, but many participants never get to find them out.

- There is little evidence to guide researchers on how best to share results with the people taking part in their trials.

### What did the researchers do and find?

- We carried out a study to test different ways of sharing trial results with participants in an ovarian cancer trial.

- We randomly assigned hospitals that were part of the ovarian cancer trial to share results with the women taking part in different ways: a basic webpage or an enhanced webpage; a printed summary of the results by mail; and an email list to receive the results.

- Nine in 10 women wanted to be told the results of the trial they had taken part in.

- Women at hospitals which sent out the printed summary by mail, were more likely to be satisfied with how the results were shared and were more likely to find out the results than those at other hospitals.

- Women who received the results said that the information was easy to understand and find, were glad and did not regret finding out the results.

## What do these findings mean?

- These findings suggest that trials with similar participants to our ovarian cancer trial (mainly women aged 50 or older), where webpages are used to share results with people taking part, should also share results through opt-out mailed printed summaries.

- This will enable more people who want to know the results to find them out, and improve satisfaction.

## Introduction

Sharing results with people who have taken part in trials is an ethical imperative [1], with the Declaration of Helsinki saying "All medical research subjects should be given the option of being informed about the general outcome and results of the study" [2]. Doing this demonstrates respect for their contribution, with some suggestion that it may increase the likelihood of participants taking part in future medical research, or recommending taking part in trials to others [3–7]. Studies have repeatedly shown that, while most participants want to receive results [3,8–12], many are not offered the opportunity to receive them [13–15].

Trial teams may be uncertain about which method to use for sharing results with participants. Most of the current evidence is based on surveys of participants or the public, prospectively asking how they would prefer to be informed, or retrospectively asking whether an approach that was used was acceptable or understandable, rather than systematically comparing outcomes from different approaches [3–6,8,11,12,14,16–21]. Most of the published evidence to date relates to sharing results with participants via mailed letters or leaflets; these studies generally report high acceptability of this approach [3,5,6,8,9,12,14,17,18,22,23]. However, sending out results by mail has resource implications. Sharing results via webpages has a number of potential advantages, including the ability to offer links to further information or support, and include audio and visual content alongside written summaries, and being discoverable by participants who have been lost to follow-up. There are also potential drawbacks in terms of accessibility for populations with low computer literacy. Fewer studies have reported sharing results via webpages. Mancini and colleagues randomised participants in a breast cancer trial to receive a letter containing a link to a website with the trial results, or no letter. They found that participants who received the letter had better understanding of the results but were not significantly more likely to have received the trial results than participants who did not receive the letter [24]. Other studies have reported low uptake of results shared via webpages [14,18], or lower levels of satisfaction with how the results were shared [12]. There is less evidence around the use of email to share results with participants; however, some studies

have found that potential research participants would be happy to receive results that way [16]. Other approaches to sharing results that have been reported include face-to-face meetings [12,18,21], teleconferences [20], and individual telephone calls or helpline services [3,20,22], although the resource requirements for these approaches may be prohibitive, particularly to large trials, and uptake of these services may be low, with Dixon-Woods and colleagues reporting no calls to a telephone helpline [22].

ICON8 (ISRCTN10356387) was a Phase III randomised controlled trial looking at 3 chemotherapy regimens for up-front treatment of ovarian cancer. Results from the earlier of the 2 co-primary endpoints, progression-free survival, were published in 2019 [25], showing no difference in progression-free survival between the 3 regimens. The Show Results to Participants Engaged in Clinical Trials (Show RESPECT) study (ISRCTN96189403) sought to generate evidence to inform trial teams on how to share results with trial participants through a mixed methods cluster randomised factorial study within the ICON8 trial. Show RESPECT tested the following 3 hypotheses, in terms of participant satisfaction with how the results were communicated:

1. An Enhanced Webpage will be superior to a Basic Webpage;

2. A Mailed Printed Summary sent by post will be superior to no Mailed Printed Summary; and

3. An invitation to join an Email List to receive updates about the trial results will be superior to no invitation to join an Email List.

## Methods

Show RESPECT was a mixed methods study, comprised of a factorial cluster randomised controlled trial within a trial to assess multiple approaches to communicating trial results, and embedded explanatory qualitative study. The data collection period for the quantitative and qualitative components was concurrent. Patients were identified for the semistructured interviews from their questionnaire responses and contact form returned alongside the questionnaire, so interviews took place after quantitative data collection for those individuals (while quantitative data collection continued for others). This paper reports both qualitative and quantitative results from data collected from trial participants. We consider the qualitative and quantitative data to have equal weight in their contribution to addressing the research aims. The full protocol for the study is available online [26] and as **S4 Appendix**.

We also collected data from site staff, but results from that part of the study are beyond the scope of this paper.

### Ethics approval

The study obtained ethics approval from the London-Chelsea Research Ethics Committee, MREC number 18/LO/1011.

### Patient and public involvement

Substantial patient and public involvement (PPI) was carried out to design and conduct this study, including focus groups, a PPI survey, patient representation on the study steering group, and input from patient groups and individuals on the design and content of the interventions.

## Quantitative methods

**Trial design.** Show RESPECT was a cluster randomised 2 by 2 by 2 factorial trial within a host trial, ICON8, an RCT evaluating chemotherapy schedules in ovarian cancer. We randomised each United Kingdom trial site (secondary or tertiary hospital) in ICON8 that agreed to take part in the Show RESPECT study to a combination of interventions to feedback ICON8 trial results to participants, as shown in **Fig 1.** A cluster design was chosen for this study, as it was felt that implementing individual randomisation for sharing results would be impractical for sites. Each site was a cluster. Allocation to each intervention was on a 1:1 ratio.

**Interventions.** Participating sites sent all ICON8 patients at those sites a printed Patient Update Information Sheet thanking them for taking part in ICON8, reminding them of the aims of the ICON8 trial, informing them that trial results were now available, and how they could access them. This included the URL of their randomised webpage (Basic or Enhanced). The Patient Update Information Sheet told patients at sites randomised to the Mailed Printed Summary that they would be sent a Mailed Printed Summary of the results after 3 weeks and that they should let their ICON8 site team know if they did not want to be sent this. Patients at sites randomised to the Email List Invitation were given a URL to sign up to the email list. The Patient Update Information Sheet was based on guidance from the Health Research Authority on End of Study Information Sheets [27]. **S1 Table** contains a detailed description of the study interventions, and links to the Basic and Enhanced Webpages. The Patient Update Information Sheet (**S1 Appendix**), Mailed Printed Summary (**S2 Appendix**), and results emails (**S3 Appendix**) can be found in the Supporting information.

Randomisation 1—All participants in Show RESPECT received a link to either the Basic or Enhanced Webpage. The Basic Webpage contained a plain English summary of results, using

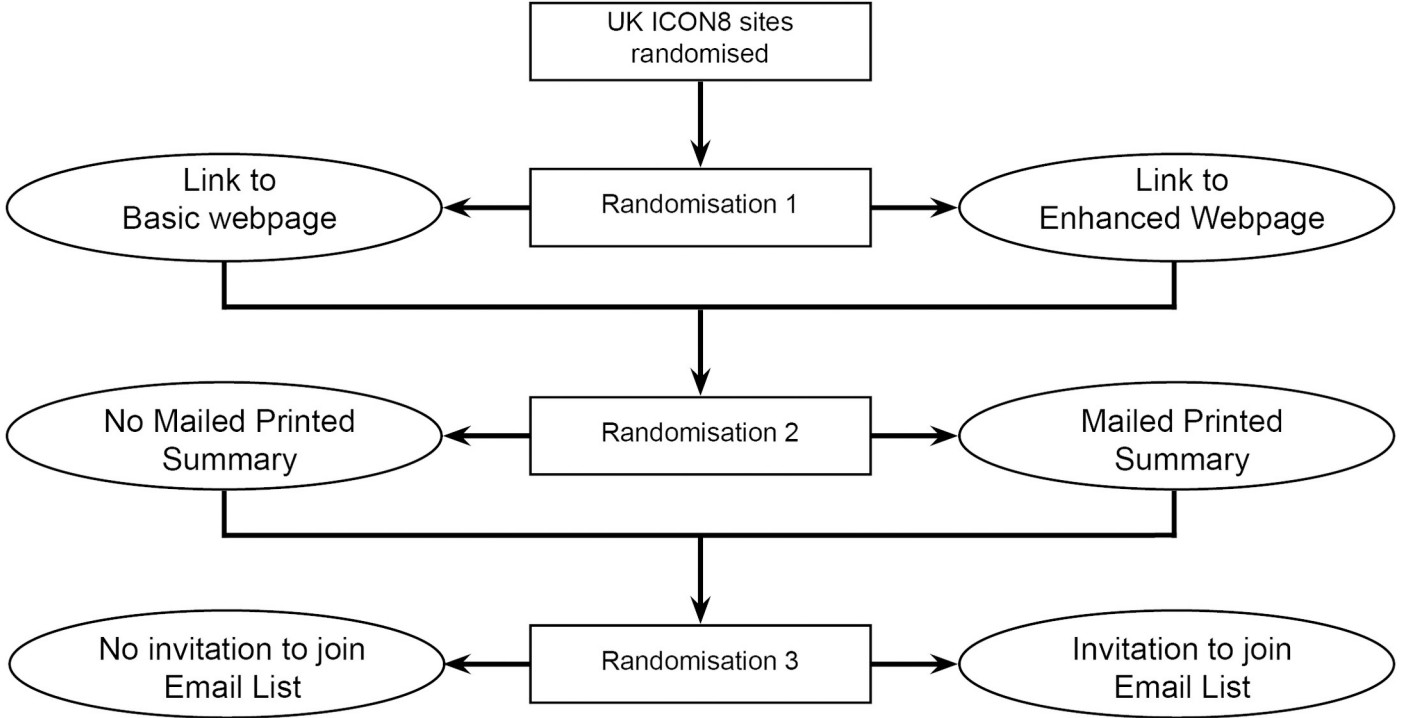

**Fig 1. Show RESPECT trial schema.** Diagram showing the 3 randomisations within Show RESPECT: (1) Link to Basic Webpage or Enhanced Webpage; (2) No Mailed Printed Summary or Mailed Printed Summary; and (3) No invitation to join Email List or Invitation to join Email List.

the structure and headings recommended for lay summaries in the EU database of clinical trials [28]. The Enhanced Webpage used a structure adapted from the Multi-Regional Clinical Trials Center guidance on feedback of results [29], and included a short video of a doctor explaining the results, links to further information and support, 2 graphics showing the trial treatment schedules and main side effects, and the opportunity to submit questions to be answered in the page's Frequently Asked Questions (FAQ) section.

Randomisation 2—The Mailed Printed Summary followed the same structure as the Enhanced Webpage, without the video or FAQ section, and was sent by post to participants' homes. The Patient Update Information Sheet for sites randomised to no Mailed Printed Summary told patients that if they were unable to access the webpage or email list, and wanted to find out the results, they should contact their research nurse.

Randomisation 3—The first email sent to those who signed up to the email list followed the same structure as the Enhanced Webpage, without the video. Participants were invited to submit questions about the results, which were answered in subsequent emails.

**Participants.** Show RESPECT collected data from women with ovarian cancer who had taken part in the ICON8 trial, were currently still alive, and in follow-up at a site participating in Show RESPECT. ICON8 participants were not invited to join Show RESPECT if they were considered by site staff to be too unwell to be contacted about this study.

**Outcomes.** Our primary outcome measure was participants' reported satisfaction with how the results were communicated to them, measured using a 5-point Likert-type scale (1 = very unsatisfied; 5 = very satisfied). Secondary outcome measures collected from participants were: the proportion of participants wanting to know the results that did find out; ease of finding out the results; whether the information about the trial results told participants everything they wanted to know; ease of understanding the results; how upsetting participants found the results; willingness to take part in a future trial; likelihood of recommending taking part in a clinical trial to friends and family; whether participants felt glad to have found out the results; and whether participants regretted finding out the results. Apart from proportion of those who wanted to know the results who found them out, these were measured using separate 5-point Likert-type scales. Quantitative data were collected from ICON8 participants by a questionnaire sent by site staff to their home address.

Data were collected between December 2018 to September 2019. Data collection finished 4 months after the final randomisation as it was felt that longer follow-up would run the risk of participants being unable to recall their experience of receiving results accurately.

**Sample size.** At trial sites, the allocated Show RESPECT intervention was offered to all eligible ICON8 participants (through the Patient Update Information Sheet). However, we did not approach all eligible participants for data collection, so as to reduce the burden on participants and staff, and because in cluster randomised trials the marginal information value of each participant declines as cluster size increases [30]. Specifically, at small sites (≤5 eligible participants), all eligible participants were invited to provide outcome data, but at medium sites (6 to 12), we aimed to collect outcome data from 6 participants, and from large sites (≥13), we aimed to collect data from 12. For medium and large sites, the individuals invited to participate were selected at random centrally. At medium and large sites, if a participant who was invited to take part chose not to, we invited the next participant from a randomly ordered, centrally held list to take part to replace the original participant, until the target number of participants at that site was reached, or no eligible participants remained.

The primary outcome measure was ordinal but for simplicity, because of lack of knowledge of its likely distribution, and to be scientifically conservative, we considered it as a binary outcome for our power calculations. We anticipated that the proportion of respondents "satisfied" without any of the research interventions would be between 20% and 80%, and in the absence

of specific prior information considered values of the Intracluster Correlation Coefficient (ICC) between 0.01 and 0.05. We considered power to detect an effect for any of the 3 interventions, for simplicity considering each in turn, i.e., effectively conducting a power calculation for each intervention assuming the other two would have no effect. We also assumed no appreciable interactions between the 3 interventions. We calculated that, based on 21 sites with and without an intervention, and an average of 4 respondents per site (172 in total), at an ICC of 0.01, we would have 80% power to detect an increase from 20% to 40%, from 50% to 71%, or from 80% to 95% in the satisfied group. Should the ICC be 0.05, then this sample size would have provided 80% power to detect an increase from 20% to 42%, 50% to 73%, or 80% to 95%. Calculations were conducted in Stata using the "power two proportions" command and assumed a coefficient of variation in cluster size of 0.6. No power calculations were made for the secondary outcomes.

**Randomisation.** Sites were randomised in blocks of 8 (the number of allocation arms available) once sites had obtained the necessary approvals. In the first phase, we randomised 3 blocks each of one site size (small, medium, and large), but, subsequently, blocks were of mixed sizes. Randomisation was conducted through random permutation within blocks.

To ensure allocation blinding, although the Show RESPECT trial statistician generated the allocations for the blocks and was aware of which clinics featured in each block, a second statistician unaware of these allocations randomly permuted the clinic names within blocks. The allocations and clinic names for each block were then matched together by a third party and revealed to the trial team. Sites were informed of their randomised allocation and sent the matching Patient Update Information Sheet.

Clusters were recruited between September 2018 to May 2019. Sites were randomised between November 2018 to May 2019. The trial was registered in February 2019, which was after some sites had been randomised, due to human resource constraints.

**Blinding.** Once randomisation had been performed, it was not possible to blind site staff to the allocation of their site. ICON8 participants were not informed that the way they were being offered the results was determined by randomisation and were not aware of the interventions being offered to participants at other sites. The questionnaire contained an embedded informed consent element, in line with the UK Health Research Authority's guidance on proportionate approaches to informed consent for self-administered questionnaire-based research [31], with completion and return of the questionnaire taken to indicate consent to use the data has been given.

**Statistical analysis.** The full statistical analysis plan can be found in **S5 Appendix**. The primary outcome measure was defined only for participants who received the ICON8 trial results, and hence analysis for this outcome was restricted to participants who reported receiving the ICON8 results. For this reason, we describe the primary analysis as following modified intention to treat (mITT). All other secondary outcomes are similarly only defined for participants who received the ICON8 results, with the exception of "report finding out the ICON8 results," which we present separately among participants who report they wanted to find the results out, and among participants who report they did not. To assess the overall effect of the intervention, it is important to interpret the results of the primary analysis alongside results concerning the possible effect of the interventions on whether participants actually found out the ICON8 results.

In the ICON8 setting, participants' health may be poor and may deteriorate before the Show RESPECT interventions were received or between intervention exposure and follow-up by questionnaire. Participants who died or became too sick to complete a questionnaire were not considered "eligible" for data collection or analysis and were not considered as missing data.

There was no specific prior evidence to suggest whether or not there would be interactions between the interventions. We were unable to think of a mechanism for potential interactions, so designed the trial based on the assumption that the effect of each intervention (e.g., enhanced versus basic webpage) would not be substantially affected by whether or not the participant was allocated to the other interventions. Hence, the primary analysis was of the main effects of each intervention adjusting for the others. However, for the primary outcome measure, we also tested each of the 3 two-way interactions and report the effect of each of 7 intervention combinations relative to control (Basic Webpage only). Adjustments were not made for multiple testing as we view our 3 study hypotheses as distinct, so all confidence intervals (CIs) presented are at the standard 5% significance level.

To reflect the study design, we adjusted for site size stratum, and also first phase versus later randomisation phases. All models included random effects for site. Estimates were also adjusted for age (continuous–linear), education (graduate versus not), and internet use (daily versus less).

Effect measures for the interventions are estimated and presented based on regression models. Ordinal random effects logistic regression was used for the primary and other Likert-type scale outcomes unless the proportional odds assumption was clearly violated. Consider odds ratios (ORs) in relation to each way the outcome could be dichotomised, e.g., quite unsatisfied or better versus very unsatisfied, quite or very satisfied versus neither or worse. Under the proportional odds assumption, these ORs are all equal, and their common value is estimated through ordinal logistic regression. The response categories were merged for the regression analysis in the event of very low reporting of one or more categories (<5% of responses). All decisions about merging response categories were taken based on an initial dataset without cluster or allocation identifiers.

For the primary outcome measure only, we conducted prespecified subgroup analyses by age group (≤70 versus ≥71 years), allocated arm of the ICON8 trial, education category (graduate versus not), and reported internet use (daily versus not). For each subgroup analysis, the effect of each intervention within subgroups were presented, and an interaction test was conducted. All interactions were with binary subgroups, with the exception of age, which was used as a continuous variable. These subgroup analyses were conducted for each of the 3 interventions separately.

Statistical analyses were performed using Stata version 16.1 (Stata Corp, Texas).

## Qualitative methods

**Qualitative data collection.**   Semistructured interviews were carried out with participants either face-to-face (at the participant's home or other location chosen by them) or by telephone by the lead qualitative researcher, AS, who holds an MPhil and MSc, is a research communication specialist, is female, and has been trained in qualitative research methods. The interviews were informed by a topic guide (**S1 Text**), which was informed by PPI. The interviewer is a research communicator by profession and was involved in developing the interventions tested in Show RESPECT. The topic guide was amended as interviews proceeded to follow-up on issues that emerged in early interviews and to improve clarity [32]. Only the participant and interviewer were present during the interviews. No repeat interviews were carried out. Interviews lasted between 32 minutes to 102 minutes. The interviews were audio recorded, and field notes were made immediately after the interviews. The interviews were transcribed verbatim. The transcriptions were checked back against the recordings for accuracy, and any identifying data were redacted. Transcripts were not returned to participants. Free-text questions within the questionnaire were also used to collect qualitative data.

**Sampling.** Invitations to take part in interviews were sent out with the Show RESPECT questionnaire, with participants asked to complete a contact details form if they wanted to find out more about the interviews and return it alongside their questionnaire. Purposive sampling was carried out, based on their questionnaire responses, to include people offered the range of Show RESPECT interventions, different levels of satisfaction with how the results were communicated, education level, internet usage, and age. Respondents who completed the contact details form and filled one or more gaps in the sampling frame were contacted by telephone with more information about the study and, if they were willing to take part, a time and date was arranged for the interview. Participants gave written informed consent. Interviews were carried out until all the gaps in the sampling frame were filled, or until no more volunteers were available who would fill a gap in the sampling frame. Using the Information Power model [33] to assess the necessary sample size, the study aim was reasonably narrow, focusing on just one aspect of trial experience (receiving results), although it did look at several approaches to results communication. The sample specificity was dense, with all interviewees having highly relevant experiences. As described in the analysis section below, an established model was applied during the analysis. The quality of dialogue in most interviews was good, resulting in a rich dataset. The analysis strategy was cross-case. Taken together, these factors suggest that a moderate sample size should provide sufficient information power to meet the aims of the study.

**Qualitative analysis.** The first step of analysis was familiarisation with the data, by listening to the recordings and reading the transcripts a number of times, recording ideas for initial codes. A thematic analysis approach was employed [34]. Both inductive and deductive approaches were for coding the data, which was carried out by AS. Initial codes were then grouped into potential themes. Emerging themes were discussed with staff from the ICON8 and Show RESPECT trial management teams. As analysis proceeded, it was found that the Information-seeking and Communication Model (ISCM) [35] fitted the codes well, so codes were categorised using concepts from that model. The ISCM is a model of information behaviour that covers both information users and information providers, their contexts, the activities of information seeking, information use and communication, and the factors that affect them [36]. Network diagrams were produced to visualise links between codes within themes, and themes were reviewed in relation to the coded extracts. As themes were generated, we searched for cases which did not fit the existing structure. Inductive thematic saturation was reached at the 13th interview, as was data saturation. Participant checking did not take place, but a PPI discussion group was held to reflect on the emerging findings and interpretation. Analysis was conducted in Atlas.ti version 8.4 (ATLAS.ti Scientific Software Development GmbH).

A "following the thread" approach was used to triangulate the results of the qualitative and quantitative components of the study [37]. This was done at the analysis stage. Each data set was initially analysed using approaches applicable to the type of data to identify key themes and questions. The qualitative data were then interrogated to explore issues raised in the quantitative data (following the "thread" from one dataset to the other).

## Results

### Participation in Show RESPECT

**Fig 2** shows the CONSORT diagram for the study. The 83 ICON8 sites in the UK were assessed for eligibility. Approximately 40 sites were excluded for reasons including lack of ICON8 participants eligible for Show RESPECT (5), lack of capacity (6), declining to take part (4), failing to obtain site approvals in time (12), or nonresponse to the invitation (13). About

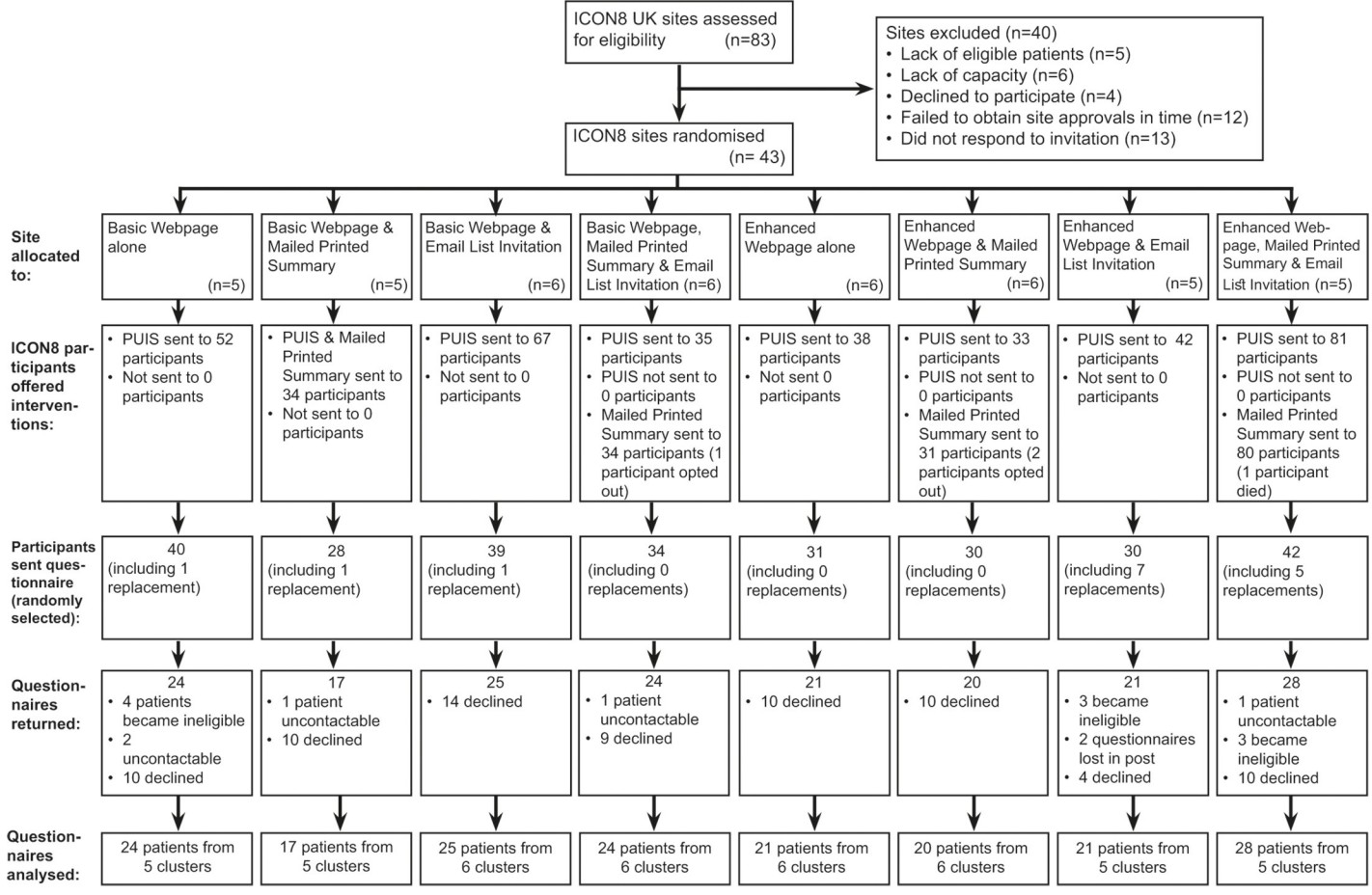

**Fig 2. CONSORT diagram for Show RESPECT.** CONSORT diagram showing flow of sites and participants through the Show RESPECT study.

43 (52%) ICON8 UK sites took part in Show RESPECT. **Table 1** shows the number of sites randomised to the interventions by site size strata, number of eligible participants who were offered the interventions, sent the questionnaire, and returned the questionnaire. Data collection took place between December 2018 and September 2019. In total, 384 ICON8 participants were offered the Show RESPECT interventions; 275 were sent the questionnaire of which 182 questionnaires were returned from 180 participants (65%) (2 of the 182 returned questionnaires were duplicates so not analysed).

**Delivery of the interventions.** Logs kept by sites showed that Patient Update Information Sheets went to 100% of eligible ICON8 participants at participating sites. Three ICON8 participants opted out of receiving the Mailed Printed Summary. According to site logs, all other eligible ICON8 participants at sites randomised to Mailed Printed Summaries were sent them.

**Baseline characteristics of participants.** **Table 2** shows the baseline characteristics of those who returned the questionnaire, and **S2 Table** shows the baseline characteristics of all eligible ICON8 participants at Show RESPECT sites. The mean age of participants who returned the questionnaire was 67, with approximately one-third from each of the 3 ICON8 arms. There was a wide range of reported highest level of educational attainment, with 38 (21%) reporting no qualifications, and 41 (23%) holding a degree or higher qualification. Nearly all participants who returned the questionnaire reported English being their first

**Table 1. Recruitment of sites and participants to Show RESPECT, by site size[1].**

| | Overall | Webpage | | Mailed Printed Summary | | Email List Invitation | |
|---|---|---|---|---|---|---|---|
| | n (%) | Basic Webpage n (%) | Enhanced Webpage n (%) | No Mailed Printed Summary n (%) | Mailed Printed Summary n (%) | No Invitation n (%) | Invitation n (%) |
| **Number of sites** | | | | | | | |
| TOTAL | **43 (100)** | 22 | 21 | 21 | 22 | 21 | 22 |
| Small sites | 17 (40) | 8 (36) | 9 (43) | 8 (38) | 9 (41) | 9 (43) | 8 (36) |
| Medium sites | 13 (30) | 7 (32) | 6 (29) | 6 (29) | 7 (32) | 6 (29) | 7 (32) |
| Large sites | 13 (30) | 7 (32) | 6 (29) | 7 (33) | 6 (27) | 6 (29) | 7 (32) |
| **Number of eligible participants (offered interventions)** | | | | | | | |
| TOTAL | **384** | 190 | 194 | 201 | 183 | 157 | 227 |
| Small sites | 54 (14) | 24 (13) | 30 (15) | 27 (13) | 27 (15) | 32 (20) | 22 (10) |
| Medium sites | 76 (20) | 45 (24) | 31 (16) | 37 (18) | 39 (21) | 35 (22) | 41 (18) |
| Large sites | 254 (66) | 121 (64) | 133 (69) | 137 (68) | 117 (64) | 90 (57) | 164 (72) |
| **Number of participants sent the questionnaire** | | | | | | | |
| TOTAL | **275** | 142 | 133 | 141 | 134 | 129 | 146 |
| Small sites | 53 (19) | 24 (17) | 29 (22) | 26 (18) | 27 (20) | 31 (24) | 22 (15) |
| Medium sites | 67 (24) | 40 (28) | 27 (20) | 30 (21) | 37 (28) | 33 (26) | 34 (23) |
| Large sites | 155 (56) | 78 (55) | 77 (58) | 85 (60) | 70 (52) | 65 (50) | 90 (62) |
| **Number of participants who returned the questionnaire (number analysed)** | | | | | | | |
| TOTAL | **180** | 90 | 90 | 91 | 89 | 82 | 98 |
| Small sites | 40 (22) | 15 (17) | 25 (28) | 21 (23) | 19 (21) | 21 (26) | 19 (19) |
| Medium sites | 49 (27) | 30 (33) | 19 (21) | 23 (25) | 26 (29) | 26 (32) | 23 (23) |
| Large sites | 91 (51) | 45 (50) | 46 (51) | 47 (52) | 44 (49) | 35 (43) | 56 (57) |
| **Response rate (percent of questionnaires sent that were returned)** | | | | | | | |
| TOTAL | **65%** | 63% | 68% | 65% | 66% | 64% | 67% |
| Small sites | 75% | 63% | 86% | 81% | 70% | 68% | 86% |
| Medium sites | 73% | 75% | 70% | 77% | 70% | 79% | 68% |
| Large sites | 59% | 58% | 60% | 55% | 63% | 54% | 62% |

[1]Small sites had 5 or fewer ICON8 patients, medium sites 6–12 ICON8 patients, and large sites 13 or more ICON8 patients alive at the time of the site agreeing to be part of Show RESPECT.

language. About 61 (40%) respondents reported using the internet or email less frequently than every day, with 26 (15%) never using internet or email.

About 94 participants were invited to take part in the qualitative study, and 13 (14%) were interviewed. **Table 3** shows the characteristics of the interviewed participants. The parts of the sampling frame we were unable to recruit participants for were "opted out of Mailed Printed Summary," "had used the email list," and "aged 50 or younger."

## Primary outcome: Did the interventions improve satisfaction with how the results were shared?

**Quantitative findings on satisfaction with how the results were shared.** Tables **4–6** shows the patient-reported outcomes relating to the experience of receiving the results, by randomised intervention. For the primary outcome of participant satisfaction with how the results were communicated, among the 3 interventions, only the Mailed Printed Summary led to a significant improvement (adjusted OR 3.15, 95% CI 1.66 to 5.98, $p < 0.001$). The effect sizes for the Enhanced versus Basic Webpages (adjusted OR 1.47, 95% CI 0.78 to 2.76, $p = 0.235$) and Email List Invitation (adjusted OR 1.38, 95% CI 0.72 to 2.63, $p = 0.327$) were much

**Table 2. Baseline characteristics of participants who returned the questionnaire.**

| | Webpage | | Mailed Printed Summary | | Email List Invitation | | Overall |
|---|---|---|---|---|---|---|---|
| | Basic Webpage n (%) | Enhanced Webpage n (%) | No printed summary n (%) | Printed summary n (%) | No Invitation n (%) | Invitation n (%) | n (%) |
| **Age** | | | | | | | |
| *Mean (IQR)* | 67 (61–74) | 68 (62–74) | 67 (61–74) | 68 (62–74) | 68 (63–75) | 67 (61–73) | 67 (62–74) |
| ≤70 years | 52 (58) | 51 (57) | 52 (57) | 51 (57) | 43 (52) | 60 (61) | 103 (57) |
| >70 years | 38 (42) | 39 (43) | 39 (43) | 38 (43) | 39 (48) | 38 (39) | 77 (43) |
| **ICON8 arm** | | | | | | | |
| Standard treatment | 26 (29) | 31 (34) | 29 (32) | 28 (31) | 25 (30) | 32 (33) | 57 (32) |
| Dose fractionated paclitaxel | 33 (37) | 28 (31) | 32 (35) | 29 (33) | 28 (34) | 33 (34) | 61 (34) |
| Dose fractionated carboplatin and paclitaxel | 31 (34) | 31 (34) | 30 (33) | 32 (36) | 29 (35) | 33 (34) | 62 (34) |
| **Highest level of educational attainment** | | | | | | | |
| No qualifications | 14 (16) | 24 (27) | 25 (27) | 13 (15) | 19 (23) | 19 (20) | 38 (21) |
| GCSE or equivalent | 28 (31) | 29 (33) | 26 (29) | 31 (36) | 32 (40) | 25 (26) | 57 (32) |
| A-level or equivalent | 25 (28) | 17 (19) | 18 (20) | 24 (28) | 17 (21) | 25 (26) | 42 (24) |
| Undergraduate degree | 11 (12) | 13 (15) | 11 (12) | 13 (15) | 8 (10) | 16 (16) | 24 (13) |
| Postgraduate degree | 11 (12) | 6 (7) | 11 (12) | 6 (7) | 5 (6) | 12 (12) | 17 (10) |
| **English as first language** | | | | | | | |
| Yes | 82 (93) | 90 (100) | 85 (96) | 87 (98) | 78 (98) | 94 (96) | 172 (97) |
| No | 6 (7) | 0 (0) | 4 (4) | 2 (2) | 2 (3) | 4 (4) | 6 (3) |
| **Use of internet or email** | | | | | | | |
| Never | 17 (19) | 9 (10) | 13 (14) | 13 (15) | 11 (13) | 15 (15) | 26 (15) |
| Once per month at most | 3 (3) | 4 (4) | 4 (4) | 3 (3) | 6 (7) | 1 (1) | 7 (4) |
| More than once per month, but not as often as every week | 1 (1) | 10 (11) | 6 (7) | 5 (6) | 0 (0) | 11 (11) | 11 (6) |
| Once per week or more, but not as often as every day | 10 (11) | 17 (19) | 15 (17) | 12 (13) | 16 (20) | 11 (11) | 27 (15) |
| Every day | 58 (65) | 50 (56) | 52 (58) | 56 (63) | 49 (60) | 59 (61) | 108 (60) |

smaller. Furthermore, there was no evidence of interaction between any pair of interventions (interaction between webpage and printed summary $p = 0.161$, webpage and email $p = 0.624$, printed summary and email $p = 0.995$).

There was no evidence of heterogeneity of the effects of the interventions on the primary outcome by age, arm in ICON8, education, or reported frequency of internet or email use (**S3 Table** and **S1 Fig**). When the 8 possible combinations of interventions were looked at individually, only those that contained the Mailed Printed Summary significantly improved the odds of participants reporting being satisfied with how the results were communicated (**S4 Table**).

**Qualitative findings on the reasons for satisfaction.** **S5 Table** contains a description of the categories from the qualitative data. Participants cited many reasons for their reported satisfaction, including characteristics related to the information products (clear and understandable); ease of accessing the results; receiving results in their preferred way; the process by which they received results (the Patient Update Information Sheet being sent out first to give them options); and their reflections on the emotional impact of the results and perceived impact for others. Participants who were unsatisfied with how the results were shared (16% of questionnaire respondents) cited a number of reasons, including the following: not knowing how to find out the results; problems accessing the webpage; finding the results difficult to understand; preferring to have found out the results in a different way (for example, wanting a

**Table 3. Characteristics of qualitative interviewees.**

| Characteristics | No. of interviewees |
|---|---|
| *Total number of interviewees* | *13* |
| **Interventions offered**[1] | |
| Basic Webpage | 8 |
| Enhanced Webpage | 5 |
| Mailed Printed Summary | 6 |
| No Mailed Printed Summary | 7 |
| Email List Invitation | 9 |
| No Email List Invitation | 4 |
| **Interventions used**[2] | |
| Basic Webpage | 5 |
| Enhanced Webpage | 2 |
| Mailed Printed Summary | 6 |
| Opted out of Mailed Printed Summary | 0 |
| Email list | 0 |
| Had not found out the results prior to interview | 2 |
| **Reported satisfaction with how the results were shared (from quantitative questionnaire)**[3] | |
| Very unsatisfied, quite unsatisfied, or neither satisfied nor unsatisfied | 5 |
| Quite satisfied or very satisfied | 5 |
| **Reported highest level of education**[4] | |
| A levels or lower | 6 |
| Degree or higher | 6 |
| **Reported frequency of internet/email use** | |
| Less than once a week | 2 |
| More than once a week | 11 |
| **ICON8 randomised allocation** | |
| Three-weekly chemotherapy (control arm) | 3 |
| Weekly chemotherapy (ICON8 intervention arms) | 10 |
| **Age group** | |
| $\leq 50$ | 0 |
| 51–60 | 2 |
| 61–70 | 6 |
| $\geq 71$ | 5 |

[1]Adds up to >13 as some participants were offered more than one intervention.

[2]Adds up to >13 as some participants used more than one intervention.

[3]Data missing from 3 participants' questionnaires.

[4]Data missing from 1 participant's questionnaire.

more personal approach, such as being told the results in person or by telephone); perceived lack of timeliness in receiving the results; and the information not giving enough detail.

## Did patients want the results, and did they find them out?

Nearly all participants (164/177 (93%)) reported wanting to know the results, and 145 (88%) of these 164 reported finding out the results. None of the participants who reported not wanting to know the results reported having found them out. These 13 participants were spread across the Show RESPECT interventions.

**Table 4. Reported outcomes relating to experience of receiving the results by randomised intervention: Enhanced versus Basic Webpage.**

| | Basic Webpage n. (%) | Enhanced Webpage n. (%) | uOR[1] (95% CI) p-value | aOR[2] (95% CI) p-value | Overall n (%) |
|---|---|---|---|---|---|
| **Proportion of patients who reported finding out the results** | | *Number for whom data were available for this outcome:* | | | 180 |
| Wanted results | 71 (89) | 74 (88) | 0.85 (0.31 to 2.32) p = 0.753 | 0.91 (0.33 to 2.54) p = 0.864 | 145 (88) |
| Did not want results | 0 | 0 | | | 0 (0) |
| **Reported satisfaction with how the results were communicated (primary outcome)** | | *Number for whom data were available for this outcome:* | | | 145 |
| Very unsatisfied | 6 (9) | 6 (8) | 1.39 (0.75 to 2.59) p = 0.295 | 1.47 (0.78 to 2.76) p = 0.235 | 12 (8) |
| Quite unsatisfied | 8 (12) | 4 (5) | | | 12 (8) |
| Neither | 11 (16) | 6 (8) | | | 17 (12) |
| Quite satisfied | 16 (23) | 24 (32) | | | 40 (28) |
| Very satisfied | 28 (41) | 36 (47) | | | 64 (44) |
| **The information told me everything I wanted to know[3]** | | *Number for whom data were available for this outcome:* | | | 146 |
| Strongly disagree | 0 (0) | 3 (4) | **2.13 (1.13 to 4.00) p = 0.019** | **2.15 (1.13 to 4.07) p = 0.019** | 3 (2) |
| Slightly disagree | 5 (7) | 2 (3) | | | 7 (5) |
| Neither | 16 (23) | 10 (13) | | | 26 (18) |
| Slightly agree | 21 (30) | 13 (17) | | | 34 (23) |
| Strongly agree | 28 (40) | 48 (63) | | | 76 (52) |
| **The information was easy to understand[3]** | | *Number for whom data were available for this outcome:* | | | 146 |
| Strongly disagree | 2 (3) | 4 (5) | 0.92 (0.47 to 1.81) p = 0.817 | 1.05 (0.53 to 2.08) p = 0.895 | 6 (4) |
| Slightly disagree | 4 (6) | 1 (1) | | | 5 (3) |
| Neither | 10 (14) | 8 (11) | | | 18 (12) |
| Slightly agree | 10 (14) | 16 (21) | | | 26 (18) |
| Strongly agree | 44 (63) | 47 (62) | | | 91 (62) |
| **It was easy to find the trial results** | | *Number for whom data were available for this outcome:* | | | 144 |
| Strongly disagree | 5 (7) | 3 (4) | 1.34 (0.71 to 2.53) p = 0.373 | 1.75 (0.90 to 3.42) p = 0.100 | 8 (6) |
| Slightly disagree | 5 (7) | 4 (5) | | | 9 (6) |
| Neither | 14 (21) | 7 (9) | | | 21 (15) |
| Slightly agree | 8 (12) | 19 (25) | | | 27 (19) |
| Strongly agree | 36 (53) | 43 (57) | | | 79 (55) |
| **I am glad I found out the trial results[4]** | | *Number for whom data were available for this outcome:* | | | 145 |
| Strongly disagree | 0 (0) | 2 (3) | 0.79 (0.38 to 1.65) p = 0.533 | 0.84 (0.40 to 1.75) p = 0.638 | 2 (1) |
| Slightly disagree | 1 (1) | 1 (1) | | | 2 (1) |
| Neither | 7 (10) | 7 (9) | | | 14 (10) |
| Slightly agree | 12 (17) | 13 (17) | | | 25 (17) |
| Strongly agree | 50 (71) | 52 (69) | | | 102 (70) |
| **I regret finding out the trial results[5]** | | *Number for whom data were available for this outcome:* | | | 138 |
| Strongly disagree | 53 (79) | 48 (68) | 1.51 (0.74 to 3.01) p = 0.253 | 1.41 (0.68 to 2.92) p = 0.354 | 101 (73) |
| Slightly disagree | 3 (4) | 9 (13) | | | 12 (9) |
| Neither | 9 (13) | 12 (17) | | | 21 (15) |
| Slightly agree | 2 (3) | 1 (1) | | | 3 (2) |
| Strongly agree | 0 (0) | 1 (1) | | | 1 (1) |
| **I found the results upsetting** | | *Number for whom data were available for this outcome:* | | | 140 |
| Strongly disagree | 40 (59) | 35 (49) | 1.26 (0.66 to 2.41) p = 0.485 | 1.24 (0.65 to 2.39) p = 0.514 | 75 (54) |
| Slightly disagree | 5 (7) | 7 (10) | | | 12 (9) |
| Neither agree nor disagree | 11 (16) | 19 (26) | | | 30 (21) |
| Slightly agree | 7 (10) | 9 (13) | | | 16 (11) |
| Strongly agree | 5 (7) | 2 (3) | | | 7 (5) |

[1]Adjusted for strata, randomisation phase (early vs late), and clustering.

[2]Adjusted for age, education level, and internet use as well as strata, randomisation phase (early vs late), and clustering.

[3]For producing the ORs for this variable, the strongly and slightly disagree categories were merged.

[4]For calculating the ORs, the strongly disagree, slightly disagree, and neither agree nor disagree categories were merged for this variable.

[5]For calculating the ORs, the neither agree nor disagree, slightly agree, and strongly agree categories were merged for this variable.

aOR, adjusted OR; OR, odds ratio; uOR, unadjusted OR.

**Table 5. Reported outcomes relating to experience of receiving the results by randomised intervention: Mailed Printed Summary versus no Mailed Printed Summary.**

| | No Mailed Printed Summary n. (%) | Mailed Printed Summary n.(%) | uOR[1] (95% CI) p-value | aOR[2] (95% CI) p-value | Overall n (%) |
|---|---|---|---|---|---|
| **Proportion of patients who reported finding out the results** | *Number for whom data were available for this outcome:* | | | | *180* |
| Wanted results | 67 (83) | 78 (94) | **3.27 (1.10 to 9.70) p = 0.032** | **3.57 (1.18 to 10.77) p = 0.024** | 145 (88) |
| Did not want results | 0. | 0 | | | 0 (0) |
| **Reported satisfaction with how the results were communicated (primary outcome)** | *Number for whom data were available for this outcome:* | | | | *145* |
| Very unsatisfied | 6 (9) | 6 (7) | **3.27 (1.74 to 6.16) p < 0.001** | **3.15 (1.66 to 5.98) p < 0.001** | 12 (8) |
| Quite unsatisfied | 7 (11) | 5 (6) | | | 12 (8) |
| Neither | 12 (19) | 5 (6) | | | 17 (12) |
| Quite satisfied | 23 (36) | 17 (21) | | | 40 (28) |
| Very satisfied | 16 (25) | 48 (59) | | | 64 (44) |
| **The information told me everything I wanted to know[3]** | *Number for whom data were available for this outcome:* | | | | *146* |
| Strongly disagree | 0 (0) | 3 (4) | 1.32 (0.70 to 2.46) p = 0.391 | 1.32 (0.70 to 2.48) p = 0.394 | 3 (2) |
| Slightly disagree | 1 (2) | 6 (7) | | | 7 (5) |
| Neither | 15 (23) | 11 (14) | | | 26 (18) |
| Slightly agree | 20 (31) | 14 (17) | | | 34 (23) |
| Strongly agree | 29 (45) | 47 (58) | | | 76 (52) |
| **The information was easy to understand[4]** | *Number for whom data were available for this outcome:* | | | | *146* |
| Strongly disagree | 2 (3) | 4 (5) | 1.60 (0.82 to 3.11) p = 0.167 | 1.66 (0.84 to 3.27) p = 0.144 | 6 (4) |
| Slightly disagree | 4 (6) | 1 (1) | | | 5 (3) |
| Neither | 10 (14) | 8 (11) | | | 18 (12) |
| Slightly agree | 10 (14) | 16 (21) | | | 26 (18) |
| Strongly agree | 44 (63) | 47 (62) | | | 91 (62) |
| **It was easy to find the trial results** | *Number for whom data were available for this outcome:* | | | | *144* |
| Strongly disagree | 3 (5) | 5 (6) | 1.15 (0.61 to 2.18) p = 0.662 | 1.37 (0.71 to 2.66) p = 0.345 | 8 (6) |
| Slightly disagree | 7 (11) | 2 (3) | | | 9 (6) |
| Neither | 6 (9) | 15 (19) | | | 21 (15) |
| Slightly agree | 14 (22) | 13 (16) | | | 27 (19) |
| Strongly agree | 34 (53) | 45 (56) | | | 79 (55) |
| **I am glad I found out the trial results[5]** | *Number for whom data were available for this outcome:* | | | | *145* |
| Strongly disagree | 0 (0) | 2 (3) | 1.69 (0.81 to 3.50) p = 0.161 | 1.69 (0.81 to 3.53) p = 0.162 | 2 (1) |
| Slightly disagree | 0 (0) | 2 (3) | | | 2 (1) |
| Neither | 9 (14) | 5 (6) | | | 14 (10) |
| Slightly agree | 14 (21) | 11 (14) | | | 25 (17) |
| Strongly agree | 43 (65) | 59 (75) | | | 102 (70) |
| **I regret finding out the trial results[6]** | *Number for whom data were available for this outcome:* | | | | *138* |
| Strongly disagree | 45 (70) | 56 (76) | 0.93 (0.46 to 1.88) p = 0.850 | 0.94 (0.46 to 1.91) p = 0.856 | 101 (73) |
| Slightly disagree | 7 (11) | 5 (7) | | | 12 (9) |
| Neither | 10 (16) | 11 (15) | | | 21 (15) |
| Slightly agree | 2 (3) | 1 (1) | | | 3 (2) |
| Strongly agree | 0 (0) | 1 (1) | | | 1 (1) |
| **I found the results upsetting** | *Number for whom data were available for this outcome:* | | | | *140* |
| Strongly disagree | 35 (55) | 40 (53) | 1.21 (0.64 to 2.30) p = 0.564 | 1.31 (0.68 to 2.51) p = 0.421 | 75 (54) |
| Slightly disagree | 6 (9) | 6 (8) | | | 12 (9) |
| Neither agree nor disagree | 15 (23) | 15 (20) | | | 30 (21) |
| Slightly agree | 8 (13) | 8 (11) | | | 16 (11) |
| Strongly agree | 0 (0) | 7 (9) | | | 7 (5) |

[1]Adjusted for strata, randomisation phase (early vs late), and clustering.

[2]Adjusted for age, education level, and internet use as well as strata, randomisation phase (early vs late), and clustering.

[3]For producing the ORs for this variable, the strongly and slightly disagree categories were merged.

[4]For producing the ORs for this variable, the strongly and slightly disagree categories were merged.

[5]For calculating the ORs, the strongly disagree, slightly disagree, and neither agree nor disagree categories were merged for this variable.

[6]For calculating the ORs, the neither agree nor disagree, slightly agree, and strongly agree categories were merged for this variable.

aOR, adjusted OR; OR, odds ratio; uOR, unadjusted OR.

**Table 6. Reported outcomes relating to experience of receiving the results by randomised intervention: Email List Invitation versus no Email List Invitation.**

| | No Email List Invitation n. (%) | Email List Invitation n. (%) | uOR[1] (95% CI) p-value | aOR[2] (95% CI) p-value | Overall n (%) |
|---|---|---|---|---|---|
| **Proportion of patients who reported finding out the results** | *Number for whom data were available for this outcome:* | | | | *180* |
| Wanted results | 65 (88) | 80 (89) | 0.96 (0.35 to 2.61) p = 0.935 | 0.78 (0.27 to 2.22) p = 0.641 | 145 (88) |
| Did not want results | 0 | 0 | | | 0 (0) |
| **Reported satisfaction with how the results were communicated (primary outcome)** | *Number for whom data were available for this outcome:* | | | | *145* |
| Very unsatisfied | 8 (12) | 4 (5) | 1.33 (0.71 to 2.47) p = 0.373 | 1.38 (0.72 to 2.63) p = 0.327 | 12 (8) |
| Quite unsatisfied | 8 (12) | 4 (5) | | | 12 (8) |
| Neither | 8 (12) | 9 (11) | | | 17 (12) |
| Quite satisfied | 13 (20) | 27 (34) | | | 40 (28) |
| Very satisfied | 29 (44) | 35 (44) | | | 64 (44) |
| **The information told me everything I wanted to know[3]** | *Number for whom data were available for this outcome:* | | | | *146* |
| Strongly disagree | 1 (1) | 2 (3) | 1.12 (0.60 to 2.10) p = 0.728 | 1.11 (0.58 to 2.12) p = 0.759 | 3 (2) |
| Slightly disagree | 3 (4) | 4 (5) | | | 7 (5) |
| Neither | 13 (19) | 13 (16) | | | 26 (18) |
| Slightly agree | 16 (24) | 18 (23) | | | 34 (23) |
| Strongly agree | 34 (51) | 42 (53) | | | 76 (52) |
| **The information was easy to understand[3]** | *Number for whom data were available for this outcome:* | | | | *146* |
| Strongly disagree | 2 (3) | 4 (5) | 0.85 (0.43 to 1.66) p = 0.627 | 0.79 (0.39 to 1.59) p = 0.500 | 6 (4) |
| Slightly disagree | 3 (4) | 2 (3) | | | 5 (3) |
| Neither | 8 (12) | 10 (13) | | | 18 (12) |
| Slightly agree | 10 (15) | 16 (20) | | | 26 (18) |
| Strongly agree | 44 (66) | 47 (59) | | | 91 (62) |
| **It was easy to find the trial results** | *Number for whom data were available for this outcome:* | | | | *144* |
| Strongly disagree | 5 (8) | 3 (4) | 0.81 (0.42 to 1.54) p = 0.511 | 0.70 (0.36 to 1.38) p = 0.306 | 8 (6) |
| Slightly disagree | 4 (6) | 5 (6) | | | 9 (6) |
| Neither | 6 (9) | 15 (19) | | | 21 (15) |
| Slightly agree | 11 (17) | 16 (20) | | | 27 (19) |
| Strongly agree | 39 (60) | 40 (51) | | | 79 (55) |
| **I am glad I found out the trial results[4]** | *Number for whom data were available for this outcome:* | | | | *145* |
| Strongly disagree | 1 (2) | 1 (1) | 0.80 (0.39 to 1.67) p = 0.555 | 0.76 (0.36 to 1.62) p = 0.475 | 2 (1) |
| Slightly disagree | 0 (0) | 2 (3) | | | 2 (1) |
| Neither | 5 (8) | 9 (11) | | | 14 (10) |
| Slightly agree | 13 (20) | 12 (15) | | | 25 (17) |
| Strongly agree | 47 (71) | 55 (70) | | | 102 (70) |
| **I regret finding out the trial results[5]** | *Number for whom data were available for this outcome:* | | | | *138* |
| Strongly disagree | 48 (76) | 53 (71) | 1.51 (0.74 to 3.08) p = 0.253 | 1.51 (0.72 to 3.16) p = 0.279 | 101 (73) |
| Slightly disagree | 7 (11) | 5 (7) | | | 12 (9) |
| Neither | 8 (13) | 13 (17) | | | 21 (15) |
| Slightly agree | 0 (0) | 3 (4) | | | 3 (2) |
| Strongly agree | 0 (0) | 1 (1) | | | 1 (1) |
| **I found the results upsetting** | *Number for whom data were available for this outcome:* | | | | *140* |
| Strongly disagree | 39 (61) | 36 (47) | 1.68 (0.87 to 3.23) p = 0.123 | 1.54 (0.79 to 3.00) p = 0.206 | 75 (54) |
| Slightly disagree | 4 (6) | 8 (11) | | | 12 (9) |
| Neither agree nor disagree | 14 (22) | 16 (21) | | | 30 (21) |
| Slightly agree | 2 (3) | 14 (18) | | | 16 (11) |
| Strongly agree | 5 (8) | 2 (3) | | | 7 (5) |

[1] Adjusted for strata, randomisation phase (early vs late), and clustering.

[2] Adjusted for age, education level, and internet use as well as strata, randomisation phase (early vs late), and clustering.

[3] For producing the ORs for this variable, the strongly and slightly disagree categories were merged.

[4] For calculating the ORs, the strongly disagree, slightly disagree, and neither agree nor disagree categories were merged for this variable.

[5] For calculating the ORs, the neither agree nor disagree, slightly agree, and strongly agree categories were merged for this variable.

aOR, adjusted OR; OR, odds ratio; uOR, unadjusted OR.

Tables **4**–**6** shows the OR and CIs by intervention, and **S6 Table** gives details of those who reported finding out the results by randomised intervention and subgroup. Of the 3 interventions, only the Mailed Printed Summary significantly increased the odds of finding out the results among those who wanted to know the results, with 78/83 (94%) reporting finding out the results, compared to 67/81 (83%) of those in the no Mailed Printed Summary arms, an OR of 3.57 (95% CI 1.18 to 10.77, *p* = 0.024), adjusted for age, education level, internet use, strata, randomisation phase, and clustering. No participants subscribed to the email list. Further information on the uptake of the interventions can be found in **S2 Text**, and qualitative findings around participants' desire for the results can be found in **S3 Text**.

## Did the information tell participants everything they wanted to know?

Most participants agreed that the information told them everything they wanted to know (**Tables 4**–**6**). Participants at sites allocated to the Enhanced Webpage were more likely to agree that the information told them everything they wanted to know (adjusted OR 2.15, 95% CI 1.13 to 4.07, *p* = 0.019) than those allocated to the Basic Webpage. There were no significant differences between the Mailed Printed Summary versus No Mailed Printed Summary, or Email List Invitation versus No Email List Invitation. See **S3 Text** for qualitative findings relating to this outcome.

## Was the information understandable?

Approximately 80% of participants reported that they found the results easy to understand. There was no statistically significant difference in any of the randomised comparisons for this outcome (**Tables 4**–**6**). See **S3 Text** for qualitative findings relating to this outcome.

## Was the information easy to find?

**Quantitative results on whether the information was easy to find.** Almost three-quarters of participants reported easily finding the results, with no significant differences between any of the Show RESPECT interventions for this outcome (**Tables 4**–**6**).

**Qualitative findings on whether the information was easy to find.** The Mailed Printed Summaries were seen as accessible to everyone, as they were not reliant on people's computer literacy or access to the internet.

*"Like my mum, for instance, in her 80s, she wouldn't have access to this [webpage], so she would only want . . . She would only be able to have posted results, really."* GMI02

When asked whether there were other ways in which they would have liked to have received the results, 22/91 (24%) questionnaire respondents from hospitals not randomised to the Mailed Printed Summary said they would have liked to receive the results by mail, with mail being seen as convenient and easier to access.

Rarely, questionnaire respondents reported not having been told how to access the results. It is unclear whether or not they received the Patient Update Information Sheet (which site logs record as having been sent). Others (from sites not randomised to the Mailed Printed Summary) reported receiving the Patient Update Information Sheet but missed the information on how to obtain the results that the sheet contained. About 11/180 questionnaire respondents reported difficulties accessing the webpage, either not having access to computers, or finding it hard to get onto the webpage, with some participants eventually gaining access, alone or with the help of family members, and others not succeeding. One woman decided to not try to access the results if it meant going online. Other participants, who had been able to

access the results themselves, were concerned that sharing results via webpages/email alone would be inaccessible to other participants, either because of lack of computer literacy or lack of access to the internet.

*"We live in quite a small community here in [County] but there's several people that aren't computer literate. And I think to presume that everybody has got access to web pages and what have you would be a mistake. And also, even things like the bandwidth or whatever you call it here is dire. Sometimes our connection is awful and I still know people in [County] who can't get a connection so if they're going to have to go to Costa Coffee to get connected to find out the results of a trial, that doesn't feel very comfortable."* DLI01

One patient commented that the process of having to type in a URL from the Patient Update Information Sheet to get to the webpage was a barrier to accessing the results, and she would have preferred to be sent them by email without having to visit a webpage to sign up for the email list.

### How did patients react to finding out the results?

While 127/145 (88%) of participants reported being glad they had found out results, only 4/138 (3%) reported regretting finding the results. About 23/140 (16%) of participants strongly or slightly agreed that they found the results upsetting, which is higher than the proportion regretting finding out the results, suggesting that while some participants were upset by the results, they did not regret having received them. There was no statistically significant difference between the arms on any of these outcomes (**Tables 4–6**). See **S3 Text** for qualitative findings around participants' emotional responses to the results.

### What did patients think about the communication interventions?

**S7 Table** summarises the qualitative feedback from questionnaires and interviews on the interventions tested within Show RESPECT.

### What were patients' attitudes to trial participation and the ICON8 results?

With no evidence of difference between the Show RESPECT interventions (**Table 7**), 131/146 (90%) of respondents reported being willing to take part in future research, and 132/147 (90%) said they were likely to recommend taking part in research to others. See **S3 Text** for qualitative findings around participants' attitudes to the research.

## Discussion

The Show RESPECT study demonstrated that sharing results with trial participants via Mailed Printed Summaries in addition to webpages increased participant satisfaction with how the results were communicated compared to webpages alone, and also enabled more participants who wanted to know the results to find them out. This satisfaction was due to the clear and understandable nature of the results summaries, ease of access, using their preferred approach, the two-stage process used, and the perceived impact of the trial (despite its "negative" results). Among women taking part in an ovarian cancer treatment trial, nearly all wanted to know the overall trial results. None of the participants who did not want to know the results found them out. It is important to look at these outcomes (satisfaction among those who received results, and proportions of people who wanted or did not want the results who received them) together, as they may not necessarily have pointed in the same direction. The two-stage

**Table 7. Reported outcomes relating to take part in or recommend taking part in research.**

| | Webpage | | | Mailed Printed Summary (MPS) | | | Email List Invitation | | | Overall n (%) |
|---|---|---|---|---|---|---|---|---|---|---|
| | Basic Webpage n (%) | Enhanced Webpage n (%) | uOR[1] (95% CI) p-value aOR[2] (95% CI) p-value | No MPS n (%) | MPS n (%) | uOR[3] (95% CI) p-value aOR[4] (95% CI) p-value | No Invitation n (%) | Invitation n (%) | uOR[3] (95% CI) p-value aOR[4] (95% CI) p-value | |
| **How willing are you to take part in future research?[5]** | | | | *Number for whom data were available for this outcome*: | | | | | | *146* |
| Very unwilling | 1 (1) | 2 (3) | uOR: 0.77 (0.37 to 1.62) $p$ = 0.494 aOR: 0.80 (0.38 to 1.70) $p$ = 0.567 | 3 (5) | 0 (0) | uOR: 1.11 (0.54 to 2.30) $p$ = 0.777 aOR: 1.09 (0.52 to 2.28) $p$ = 0.827 | 2 (3) | 1 (1) | uOR: 0.72 (0.34 to 1.51) $p$ = 0.380 aOR: 0.70 (0.33 to 1.53) $p$ = 0.375 | 3 (2) |
| Quite unwilling | 1 (1) | 1 (1) | | 1 (2) | 1 (1) | | 1 (1) | 1 (1) | | 2 (1) |
| Not sure | 6 (8) | 4 (5) | | 2 (3) | 8 (10) | | 3 (4) | 7 (9) | | 10 (7) |
| Quite willing | 9 (13) | 16 (21) | | 13 (20) | 12 (15) | | 10 (15) | 15 (19) | | 25 (17) |
| Very willing | 54 (76) | 52 (69) | | 47 (71) | 59 (74) | | 51 (76) | 55 (70) | | 106 (73) |
| **How likely are you to recommend taking part in research to others?[6]** | | | | *Number for whom data were available for this outcome*: | | | | | | *147* |
| Very unlikely | 3 (4) | 3 (4) | uOR: 1.13 (0.55 to 2.31) $p$ = 0.739 aOR: 1.17 (0.56 to 2.44) $p$ = 0.671 | 5 (7) | 1 (1) | uOR: 1.28 (0.63 to 2.62) $p$ = 0.491 aOR: 1.23 (0.59 to 2.57) $p$ = 0.579 | 2 (3) | 4 (5) | uOR: 0.82 (0.40 to 1.69) $p$ = 0.594 aOR: 0.77 (0.36 to 1.65) $p$ = 0.507 | 6 (4) |
| Quite unlikely | 1 (1) | 1 (1) | | 0 (0) | 2 (3) | | 1 (1) | 1 (1) | | 2 (1) |
| Not sure | 6 (8) | 1 (1) | | 2 (3) | 5 (6) | | 4 (6) | 3 (4) | | 7 (5) |
| Quite likely | 11 (15) | 17 (23) | | 15 (22) | 13 (16) | | 11 (16) | 17 (21) | | 28 (19) |
| Very likely | 51 (71) | 53 (71) | | 45 (67) | 59 (74) | | 49 (73) | 55 (69) | | 104 (71) |

[1]Adjusted for strata, randomisation phase (early vs late), and clustering.

[2]Adjusted for age, education level, and internet use as well as strata, randomisation phase (early vs late), and clustering.

[3]Adjusted for age, education level, and internet use as well as strata, randomisation phase (early vs late), and clustering.

[4]For producing the ORs for this variable, the strongly and slightly disagree categories were merged.

[5]For calculating the ORs, the very unwilling, quite unwilling, and not sure were merged for this variable.

[6] For calculating the ORs, the very unlikely, quite unlikely, and not sure were merged for this variable.

aOR, adjusted OR; uOR, unadjusted OR.

process, informing participants that the results are available and how to access them, rather than automatically sending results out to all participants, was important to ensure that the wishes of the 7% of participants who did not want to find out the results were respected. This may be especially important in trials where the participant population may be vulnerable, or the results may be emotionally challenging for some participants. The additional features of the Enhanced Webpage did not increase satisfaction with how the results were communicated compared to the Basic Webpage but did lead to a higher proportion of participants reporting that it told them everything they wanted to know. The lack of uptake of the Email List Invitation suggests that for similar trial populations, it is not worth creating email lists at the end of the trial.

The qualitative findings show that participants liked the Mailed Printed Summary as an approach, as it was seen as more accessible for patients with limited access to the internet or computer literacy, and it also facilitated keeping the results for future reference, or showing to friends and family. The results sparked a range of responses, including both positive emotions and disappointment and upset, but there was no evidence to suggest that they were experienced as harmful by these individuals, and nearly all participants were glad to have received the results, even if some had found them upsetting.

Our trial employed a cluster randomised factorial design to assess 3 methods of sharing results with participants, allowing us to be confident that the differences observed were due to the interventions. Extensive PPI was carried out to ensure that the study was asking a question that was important to patients and that the interventions tested were appropriate. The interventions selected were designed to be easily replicable in other studies. The mixed methods approach allowed us to explore the reasons behind the quantitative results, while gaining an overall picture across the study population.

Budget constraints meant that we were unable to send questionnaires to all ICON8 participants at the participating sites. However, we used random selection of participants to avoid selection bias, and the characteristics of respondents in terms of age and ICON8 arm are similar to that of all eligible participants at trial sites. Our response rate of those invited to complete the questionnaire was 65%. This introduces uncertainty into the interpretation of our results, as we do not know how outcomes would vary between responders and nonresponders. However, our response rate is similar to that seen in other studies looking at communication of results to trial participants [8,38]. We cannot discern if there are differences between respondents and nonrespondents in other potentially relevant characteristics (e.g., education level, computer literacy); however, respondents cover the range of these characteristics, and the subgroup analysis showed no evidence of heterogeneity in effect by these subgroups. We do not have data on the ethnicity of participants in either the host ICON8 trial or the embedded Show RESPECT trial, meaning that we cannot assess whether ethnicity influences desire to know trial results, or how these results should be shared. We are also unable to explore the impact of what respondents' first language was on their experience of receiving results. Future clinical trials of treatment may wish to systematically collect this information up front to assess inclusivity [39] and facilitate embedded trials, like Show RESPECT.

This randomised controlled trial contributes to the, as yet, scant evidence base on how to communicate study results to trial participants, providing high-quality evidence to a field that is dominated by observational data, surveys asking about hypothetical scenarios, and expert opinion. Our study adds comparative data around the effectiveness of different communication approaches in practice. Our results around participants' desire for results are consistent with findings from previous studies [4]. Our participants' positive reaction to receiving trial results is also consistent with that reported by previous studies [17,18,20,22], even in the context of potentially upsetting results [23]. **Box 1** lists our recommendations on points for trialists to consider around sharing results with trial participants.

Show RESPECT was conducted within the context of an ovarian cancer treatment trial, where the population was women with an average age of 67 years and living in the UK. It is unclear how generalizable these results are to trials with different patient populations (e.g., all male or mixed, younger participants and participants likely more familiar with technology, trials studying non-life-threatening conditions, or where results are available soon after receiving trial treatment). Webpages are a low-cost communication approach and may be useful alongside printed summaries, giving opportunities to provide links to further information and support, and audiovisual content that Printed Summaries cannot provide. However, 4 in 10 of our respondents reported using the internet or email less than daily, with 15% never using them. Data from the UK Office for National Statistics in 2019 show that 10% of the UK population are classed as internet nonusers, having either never used the internet or not used it in the last 3 months [40]. Internet nonusers in the UK are more likely to be women, over the age of 65, have a disability, or be economically inactive (particularly those on long-term sick leave) [40]. Households with lower incomes are also less likely to have an internet connection [40]. While the number of internet nonusers has been declining in recent years, trials should be careful about relying on the internet or email to share results with participants if their trial population

Box 1. Recommendations based on this research.

- Trial teams should consider when planning their study how to share results with participants, taking into account:

  - the characteristics of the study population, including, but not limited to, health literacy, computer literacy, access to the internet, age, and, likely, health status;

  - the need to offer choice to participants, allowing those who want to find out the results easy access but not forcing them on those who do not want to receive them; and

  - how to make it possible for participants to keep the results, so they can refer to them in the future.

- Trial teams should adequately budget the necessary resources to fulfil their obligation to offer the results to study participants in a way that is appropriate to the study population—the lowest cost approaches (e.g., a basic webpage) may not be optimal for every study population.

- Patient and public involvement is essential for planning how to share results with participants, identifying the outcomes and study results that are important and relevant to participants, and developing the content of results summaries.

- Care is needed to ensure that the wording of results summaries is both clear to participants and sensitively written.

overlaps with some of the groups most likely to be internet nonusers. Failure to take this into account could exclude a significant proportion of participants from accessing results. If trials are to meet the ethical obligation to offer results to participants, they need to plan and budget for this in a way that is accessible to the trial population. Trials with similar patient populations should budget for Mailed Printed Summaries or ensure that alternative approaches to webpages are easily available and known to those participants for whom webpages are inaccessible. Problems accessing the internet were not the only reasons participants preferred the printed summaries; even those who use the internet daily were more likely to be satisfied with the printed summary, which made it easy for participants to file along with other trial information and to share with others.

Other trial settings may pose different challenges around sharing results; however, qualitative research from the BRACELET study, which focused on neonatal intensive care trials, found similar responses to the receipt of results among bereaved parents [41].

The ICON8 results that we were communicating (no difference between the trial regimens) may have influenced participants' reported satisfaction, interacting with the communication interventions. The results of the ICON8 trial did not come out of the qualitative research as a major reason for satisfaction or dissatisfaction with how they were shared, and most participants in the qualitative interviews understood the importance of "negative" trial results. However, our qualitative findings do raise the question of whether modified or tailored approaches would be needed to communicate results to participants in each of the trial's randomised groups if the trial had found a strong clinical difference. People on the poorer performing arm may need additional support or more personalised approaches to receiving results. All

eventualities need to be anticipated if a feedback strategy is built into a trial protocol. Research is needed to explore whether our results are reproducible in trials that do find significant benefit or harm.

Future analysis of data collected within Show RESPECT will focus on the perspective of site staff involved in sharing the results with participants, the process used for this, and resource implications of the communication approaches used. Further research is needed to explore the issue of sharing results with the relatives of trial participants who die during a trial, to see if this is something that relatives want, and if so, how it can be done without causing unnecessary distress.

## Conclusions

There is a lot of evidence that trial participants want to be offered the overall results of their trial. A common criticism is that there is not enough guidance as to how this might happen. By testing a number of approaches in a sensitive area, and finding out what is acceptable to participants, Show RESPECT moves the field forward. For the patient population of the ICON8 ovarian cancer trial, adding Mailed Printed Summaries to web-based approaches improved patient satisfaction and was better at ensuring those who wanted to know the results were able to find them out. **Box 1** contains recommendations based on this research.

## Supporting information

**S1 Text. Topic guide for qualitative interviews with patients.**
(DOCX)

**S2 Text. Further information on uptake of interventions.**
(DOCX)

**S3 Text. Further qualitative findings.**
(DOCX)

**S1 Fig. Forest plot of satisfaction with how the results were shared, by subgroup.**
(DOCX)

**S1 Appendix. Patient Update Information Sheet.**
(PDF)

**S2 Appendix. Mailed Printed Summary.**
(PDF)

**S3 Appendix. Results email sent to those who signed up to the email list.**
(PDF)

**S4 Appendix. Show RESPECT Protocol.**
(PDF)

**S5 Appendix. Statistical analysis plan.**
(PDF)

**S1 Table. Description of the interventions tested in Show RESPECT.**
(DOCX)

**S2 Table. Baseline characteristics of all eligible participants at trial sites.**
(DOCX)

**S3 Table. Subgroup analyses (primary outcome only).**
(DOCX)

**S4 Table. Effect of combinations of interventions on satisfaction with how the results were shared.**
(DOCX)

**S5 Table. Qualitative data categories.**
(DOCX)

**S6 Table. Proportion of participants who wanted to know the results reporting finding them out, by randomisation and subgroup.**
(DOCX)

**S7 Table. Qualitative feedback on the interventions within Show RESPECT.**
(DOCX)

**S8 Table. CONSORT 2010 checklist of information to include when reporting a cluster randomised trial.**
(DOCX)

**S9 Table. COREQ (COnsolidated criteria for REporting Qualitative research) Checklist.**
(PDF)

**S10 Table. List of staff involved at Show RESPECT sites.**
(DOCX)

## Acknowledgments

We are very thankful to the patients who participated in this study and the people who contributed to our Patient and Public Involvement activities. We are very grateful for the work of the Show RESPECT site teams in carrying out this study. A list of team members from sites can be found in **S10 Table**. We thank Eva Burnett, who is a patient representative on the Study Steering Group for Show RESPECT and ICON8, for her input on the design and conduct of this study and for her comments on this manuscript, and Amanda Hunn for her contributions to the steering group. We also acknowledge the hard work and diligence of Sierra Santana and Ania Spurdens, who were data managers for Show RESPECT. We also wish to thank the ICON8 trial team for their support of this project, particularly Andrew Clamp, Babasola Popoola, Francesca Schiavone, Jonathan Badrock, and Rick Kaplan. We also thank Julia Bailey for her advice and guidance on the qualitative aspect of this study.

## Author Contributions

**Conceptualization:** Annabelle South, Cara Purvis, William J. Cragg, Conor Tweed, Matthew R. Sydes, Claire Snowdon, Katie Gillies, Talia Isaacs, Barbara E. Bierer, Andrew J. Copas.

**Data curation:** Annabelle South, Cara Purvis, Carlos Diaz-Montana.

**Formal analysis:** Annabelle South, Andrew J. Copas.

**Funding acquisition:** Matthew R. Sydes.

**Investigation:** Nalinie Joharatnam-Hogan, Archie Macnair.

**Methodology:** Annabelle South, William J. Cragg.

**Project administration:** Annabelle South, Cara Purvis, Carlos Diaz-Montana.

**Software:** Carlos Diaz-Montana.

**Supervision:** Nalinie Joharatnam-Hogan, Elizabeth C. James, William J. Cragg, Conor Tweed, Archie Macnair, Matthew R. Sydes, Claire Snowdon, Katie Gillies, Talia Isaacs, Barbara E. Bierer, Andrew J. Copas.

**Validation:** Elizabeth C. James.

**Writing – original draft:** Annabelle South, Andrew J. Copas.

**Writing – review & editing:** Annabelle South, Nalinie Joharatnam-Hogan, Cara Purvis, Elizabeth C. James, Carlos Diaz-Montana, William J. Cragg, Conor Tweed, Archie Macnair, Matthew R. Sydes, Claire Snowdon, Katie Gillies, Talia Isaacs, Barbara E. Bierer, Andrew J. Copas.

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
