## [Editor Report · Decision Letter 0]

17 Jun 2021

Dear Dr South, 

Thank you for submitting your manuscript entitled "Show RESPECT: A cluster randomised, factorial, mixed methods trial testing approaches to sharing trial results with participants" for consideration by PLOS Medicine.

Your manuscript has now been evaluated by the PLOS Medicine editorial staff and I am writing to let you know that we would like to send your submission out for external peer review.

Please re-submit your manuscript within two working days, i.e. by Jun 21 2021 11:59PM.

Kind regards,

Callam Davidson

Associate Editor

PLOS Medicine

---

## [Decision Letter · Decision Letter 1]

16 Jul 2021

Dear Dr. South,

Thank you very much for submitting your manuscript "Show RESPECT: A cluster randomised, factorial, mixed methods trial testing approaches to sharing trial results with participants" (PMEDICINE-D-21-02611R1) for consideration at PLOS Medicine. 

Your paper was evaluated by an associate editor and discussed among all the editors here. It was also discussed with an academic editor with relevant expertise, and sent to independent reviewers, including a statistical reviewer. The reviews are appended at the bottom of this email and any accompanying reviewer attachments can be seen via the link below:

[LINK]

In light of these reviews, we will not be able to accept the manuscript for publication in the journal in its current form, but we would like to consider a revised version that addresses the reviewers' and editors' comments. We hope you will understand that we cannot make any decision about publication until we have seen the revised manuscript and your response, and we plan to seek re-review by one or more of the reviewers. 

We hope to receive your revised manuscript by Aug 06 2021 11:59PM. Please email us (plosmedicine@plos.org) if you have any questions or concerns.

We look forward to receiving your revised manuscript and please don't hesitate to contact me with any questions. 

Sincerely,

Callam Davidson, 

Associate Editor

PLOS Medicine

plosmedicine.org

The feeling amongst the editorial team was that the manuscript needs to be better organized and condensed before publication can be considered. Please aim to streamline the manuscript where possible, taking into account the comments from peer reviewers. One suggestion may be to separate out the presentation of the quantitative and qualitative aspects and relocate some of the qualitative work to the supplementary materials section. 

In your data availability statement, please include details of where the CTU’s ‘Data Sharing Policy’ can be located (https://www.ctu.mrc.ac.uk/our-research/other-research-policy/data-sharing).

Please revise your title according to PLOS Medicine's style. Your title must be nondeclarative and not a question. It should begin with main concept if possible. I would suggest ‘Testing approaches to sharing trial results with participants: The RESPECT cluster randomised, factorial, mixed methods trial’.

Please structure your abstract using the PLOS Medicine headings (Background, Methods and Findings, Conclusions).

Abstract Background: Provide the context of why the study is important. The final sentence should clearly state the study question.

Abstract Methods and Findings:

* Please ensure that all numbers presented in the abstract are present and identical to numbers presented in the main manuscript text (e.g. 72% is included in the abstract but I could not locate this in the main text).

* Please include the years during which the study took place and length of follow up.

* In the last sentence of the Abstract Methods and Findings section, please describe the main limitation(s) of the study's methodology."

At this stage, we ask that you include a short, non-technical Author Summary of your research to make findings accessible to a wide audience that includes both scientists and non-scientists. The Author Summary should immediately follow the Abstract in your revised manuscript (you currently include something similar but please update to match the Author Summary format). This text is subject to editorial change and should be distinct from the scientific abstract. Please see our author guidelines for more information: https://journals.plos.org/plosmedicine/s/revising-your-manuscript#loc-author-summary

Please relocate paragraph 1 (lines 406-410) of the Results section into the Methods section.

Please specify whether informed consent was written or oral.

Please begin the Discussion with a short, clear summary of the article's findings and ensure the strengths and limitations of the study are clearly described in their own paragraph later in the Discussion.

Thanks for providing a CONSORT checklist. Please update the checklist to use section and paragraph numbers, rather than page numbers (as these will change in subsequent revisions).

As part of your revision, please complete and submit a copy of the COREQ Guidelines checklist, a document that aims to improve reporting of qualitative studies for purposes of post-publication data analysis and reproducibility: https://www.equator-network.org/reporting-guidelines/coreq/

Your trial appears to have been registered after the participants were randomized. Please explain in the paper why your trial was registered late. 

The sample size listed in the submitted manuscript and the trial registry differ. This discrepancy appears to be due to the manuscript reporting participants offered the intervention as well as those that returned questionnaires. Please ensure it is made clear throughout the manuscript how many participants were included in your analyses. 

Please include the study protocol document and analysis plan, with any amendments, as Supporting Information to be published with the manuscript if accepted.

Comments from the reviewers:

Reviewer #1: Thank you for the opportunity to review this manuscript. The authors tackle an important question regarding offering research results to participants. As ICON8 was a high-impact, and large trial, I think the experience in this manuscript will be useful to investigators considering how to communicate results to participants in planned trials.

I do not think that the manuscript is suitable for publication in its current form. There is way too much text here; over 100 pages were included in the pdf for my review: the manuscript itself is ~7500 words and the tables are large and confusing to navigate. The authors' important results will not be communicated effectively if the reader (even if familiar with gynecologic cancers and communication of research results) cannot make it through the manuscript. It may be that the qualitative component should be presented in a separate submission; in my opinion the quantitative component is more important and likely to influence research practice. Suggest a target of ~3000 words for these data.

A few additional comments:

- Suggest considering using "mailed" or "sent via post" instead of "posted." For most of the early part of this paper I thought that "posted" meant "tacked to a wall in a public place" rather than "sent via postal service." 

- Description of ICON8 in lines 141-143 is confusing as written, and q 3 week chemo is not always standard of care... this is beside the point of this article and distracting to a reader familiar with the trial. Perhaps better to say RCT of three chemotherapy regimens for up-front treatment of ovarian cancer and leave it at that.

- Table 4 and according locations in the text: what do the odds ratios represent? 

- The Discussion undersells the impact of making conclusions on hard-copy versus website/email list in a population primary composed of older women. I would want to see a little more specific discussion of which populations might benefit from (cheaper) more technologically-focused methods of results communication, and which might require results sent by post.

- Would be interesting to read whether the authors believe that there is an ethical obligation to budget for sending out trial results by mail in a similar patient population.

Reviewer #2: PMEDICINE-D-21-02611R1

It is great to see a robust SWAT comparing different approaches to the dissemination of results to trial participants. Overall, I found the manuscript very well presented and most of my comments are minor.

o Major comment:

* My only major comment relates to missing data. Participants are lost at two stages: 1. a number of participants who were offered the intervention were not sent the questionnaire and 2. A number who were sent the questionnaire did not return it. I understand that only a random sample of those offered the intervention were included due to logistical constrains; however, this departs from the ITT principle. Out of the 274 participants who were sent a questionnaire, only 180 (66%) returned it. This large amount of missing data creates substantial uncertainty and would probably warrant further sensitivity analyses as well as more recognition in the discussion.

o Minor/specific comments:

* Abstract, Results: I am not sure it is necessary to provide a breakdown of returned questionnaire by randomised arm in the abstract itself.

* Abstract, Conclusion: can we really say that the study "provides *clear* evidence on how to share results". I find the wording too strong considering some of the potential limitations (e.g. missing data / potential selection biases). I would suggest rewording to "the study *some* evidence"

* I note that the webpage comparison involved an enhanced webpage compared to a basic webpage (active comparator) whereas the printed summary was compared to no summary at all. The authors conclude that the printed summary had the greatest effect on satisfaction; however, I wonder this is due to the choice of comparator. Would a fairer comparison have been webpage vs no-webpage?

* I am surprised to see that the detectable differences are almost the same with an ICC of 0.01 or 0.05 (see sample size calculation). According to my calculations, with an average cluster size of 4 patients, the inflation factor due to the cluster design goes from 1.03 to 1.15 with an ICC of 0.01 and 0.05, respectively. While this is a moderate increase, I would expect it to have more impact on the detectable difference. Please confirm/clarify.

* At the start of the analysis section (line 306), the analysis population (mITT) is defined as participants who reported receiving the ICON8 results. Should not the primary analysis population include all who were sent the printed Patient Update Information Sheet (ITT), regardless of whether they actually accessed any result? Or at least all who were sent the questionnaire? If this is a missing data issue (i.e. no satisfaction data was collected in those who did not actually receive the results), some sensitivity analyses may be required (e.g. missing data imputations).

* Table 2 shows the baseline characteristics of those who returned the questionnaire. It would be interesting to compare these to participants who did not return the questionnaire to assess the potential for biases.

* I cannot find the results of the interaction test between interventions referred to on line 460. Please consider including them.

* Please consider replacing (or augmenting) Supplementary Table 3 with a forest plot to allow an easier comparison of effects across interventions. The forest plot could potentially be included as a figure in the body of the manuscript. I note that none of the interaction tests were significant; however, these are likely underpowered and therefore unlikely to detect existing interactions. 

* Table 6 shows the results of those who reported finding out the results among those who wanted to know. Can you please clarify why the analysis is limited to those who wanted to know instead of all enrolled? Also, the text mentions odds ratios and 95% CI which do not appear in Table 6 which instead shows data by subgroups. I wonder whether data by subgroups should be in the supplement (were these pre-specified?). Please also consider adding the results of the logistic regression to Table 4 together with the other outcomes.

* Please consider shortening the results section as well as rearranging it to have all quantitative results together followed by all qualitative results.

* Please consider discussing the impact of missing data as a limitation.

* In the conclusion, I am not sure one can write that Posted Printed Summaries (PPS) were superior to web-based approaches in terms of patient satisfaction and ensuring those who wanted to know the results were able to find them out as there is no direct comparison between posted printed summaries and web-based approaches. In fact, when comparing the raw distribution between enhanced web-based and PPS, the numbers look very similar and one could argue that the two are equivalent and potentially both better than no active intervention. What we can conclude is that PPS showed better satisfaction than no PPS and that there was no significant difference between enhanced web display vs standard display

-Laurent Billot

Reviewer #3: This study takes on a very important topic: How do different methods of sharing trial results' influence trial participants' satisfaction with receiving results? The study design and implementation are very complex. Due to practical considerations related to studying participants in an existing randomized controlled trial, the study makes several necessary deviations from an idealized textbook study design and implementation. However, these limitations are outweighed by the novel approach to prospectively randomly assigning trial participants to several types of results sharing activities. These findings do add new information to what is already known, particularly by taking a random assignment approach rather than an observational/single-arm approach.

Below are some suggestions that may enhance the clarity of presentation of this complex study:

*The abstract does not immediately and clearly explain how the interventions were deployed. It would be good to make clear early on that the study is a site-level cluster randomized 2x2x2 such that participants at each site received one of eight possible combinations of communication activities.

*Because no participants actually subscribed to the email list, the "Email List" intervention might be best described as the "Email List Invitation" intervention or similar throughout the entire article and supporting materials. The abstract itself might be improved by making clear that no patients signed up for the email list.

*Line 322: This assumption is stated without explaining its justification. The reader would likely be helped by providing a brief rationale why "the prior assumption in the Show RESPECT trial design is that there will not be any important interactions between the three interventions."

*Line 500: It may be useful to clarify why there was so much missing data (40%) on the question about accessing the web pages. The quantity of missing data makes the percentages reported in the Results for this question relatively uninterpretable.

*In the Discussion, it is important to note that the specific results of the ICON8 trial may have influenced participants' satisfaction in general. The content of ICON8's specific results may have interacted with the communication intervention activities to influence participants' perceptions. This is not a major limitation, but should be clearly noted.

*In the Discussion, it may be useful to elaborate on the decision that "the primary outcome measure was defined only for participants who received the ICON8 trial results, and hence analysis for this outcome was restricted to participants who reported receiving the ICON8 results. For this reason, we describe the primary analysis as following modified intention to treat (mITT)." The main outcome could have been defined with respect to satisfaction with the overall approach by which that results sharing was handled in the trial, which would have allowed for inclusion of people who opted out or didn't receive results. From an ethical and/or funder perspective, this broader, more inclusive outcome may be at least as important as satisfaction with how results were communicated.

*Line 698: It is interesting to consider the importance of the finding that participants made a distinction between features they used vs what they thought others would like to use. This may have implications for how researchers work with participants to design results sharing activities for participants in their own studies.

*Line 721: The word "interventions" is used here to refer to the ICON8 trial, generating some confusion because "interventions" is mostly used to refer to the interventions in the results sharing trial. The authors might consider rephrasing to improve clarity.

*Line 827: The conclusion states: "Box 1 contains recommendations based on this research for trial teams, research funders and ethics committees." However, Box 1 contains advice that extends well beyond what was learned from this study. The researchers may consider removing points that are not directly supported by study findings or could make clear that these recommendations are drawn from other sources. Most of Box 1 reports broad suggestions that are not derived directly from the current study's specific findings and are available in existing resources for results sharing that could be cited in the paper rather than being reprinted here in a paper that already includes a great deal of detailed information.

[LINK]

---

## [Decision Letter · Decision Letter 2]

2 Sep 2021

Dear Dr. South,

Thank you very much for re-submitting your manuscript "Testing approaches to sharing trial results with participants: The Show RESPECT cluster randomised, factorial, mixed methods trial" (PMEDICINE-D-21-02611R2) for review by PLOS Medicine.

I have discussed the paper with my colleagues and the academic editor and it was also seen again by two reviewers. I am pleased to say that provided the remaining editorial and production issues are dealt with we are hoping to accept the paper for publication in the journal.

[LINK]

We look forward to receiving the revised manuscript by Sep 09 2021 11:59PM.   

Sincerely,

Callam Davidson, 

Associate Editor 

PLOS Medicine

cdavidson@plos.org

Requests from Editors:

Please provide legends for all figures (including those in Supporting Information files).

Thank you for providing an author summary – this needs to be trimmed slightly (each section ideally should comprise 2-3 single sentence bullet points). Suggestions for shortening are as follows:

• Please merge the bullet points on lines 94 and 96 to form a single sentence (‘Previous research has shown that most people who take part in clinical trials want to be told the results of those trials, but many participants never get to find them out’).

• Please shorten the bullet/sub-bullets beginning on line 102 to ‘We randomly assigned hospitals that were part of the ovarian cancer trial to share results with the women taking part in different ways: a basic webpage or an enhanced webpage; a printed summary of the results by mail; and an email list to receive the results.’

• Please combine the bullets on lines 111 and 113 as follows: ‘Women at hospitals which sent out the printed summary by mail were more likely to be satisfied with how the results were shared and were more likely to find out the results than those at other hospitals.’

• Please make the bullet on line 115 a single sentence (‘Women who received the results said that the information was easy to understand and find, were glad, and did not regret finding out the results.

• The bullet style appears to be different in the ‘What do these findings mean?’ section.

Lines 779-801: Please remove the ‘Data availability’ and ‘Funding source’ sections from your manuscript main body – in the event of publication, this information will be published as metadata based on your responses to the submission form. 

Data availability statement: ‘The individual participant data … will be available beginning 12 months after publication following the CTU’s standard moderated access approach’ - Apologies in advance if I have missed it, but I could not find any reference to the 12 month delay period in the link provided (https://www.ctu.mrc.ac.uk/our-research/other-research-policy/data-sharing/) - could you please provide further rationale for not making the data available on publication? 

Lines 20 – 47: Please remove the ‘Contributor and guarantor information’, ‘Copyright’ and ‘Competing interests’ sections from the main body of your manuscript. In the event of publication, author contributions and competing interests will be published as metadata based on your submission form responses (so please ensure all relevant information is captured in your answers). PLOS applies the Creative Commons Attribution (CC BY) license to articles and other works we publish – please see https://journals.plos.org/plosmedicine/s/licenses-and-copyright for more information.

Lines 90-91: Please remove the funding information and ensure it is captured in your response to the submission form.

Comments from Reviewers:

Reviewer #1: Thank you for the opportunity to re-review this manuscript. All my previous questions/comments have been addressed satisfactorily.

Reviewer #3: This manuscript has been thoroughly revised to address the editorial staff's and reviewers' comments. Each of the comments has been thoughtfully addressed in a way that improves the manuscript. While the manuscript includes extra details (e.g. describing that the lead qualitative researcher "holds an MPhil and MSC" [Line 384]) and could be shortened further, the level of detail appears to be in service of transparency and promoting reproducibility. I am unsure whether many readers will persist through the entire manuscript. However, a trial-within-a-trial requires a great deal of explanation, and this study's design and findings advance the field. It would be a good thing if other researchers see this paper and develop similar studies building off of this one.

[LINK]

---

## [Editor Report · Decision Letter 3]

7 Sep 2021

Dear Dr South, 

On behalf of my colleagues and the Academic Editor, Dr Aaron S Kesselheim, I am pleased to inform you that we have agreed to publish your manuscript "Testing approaches to sharing trial results with participants: The Show RESPECT cluster randomised, factorial, mixed methods trial" (PMEDICINE-D-21-02611R3) in PLOS Medicine.

PRESS

Sincerely, 

Callam Davidson 

Associate Editor 

PLOS Medicine